# Assessment of Technological Capabilities for Forming Al-C-B System Coatings on Steel Surfaces by Electrospark Alloying Method

**DOI:** 10.3390/ma14040739

**Published:** 2021-02-05

**Authors:** Bogdan Antoszewski, Oksana P. Gaponova, Viacheslav B. Tarelnyk, Oleksandr M. Myslyvchenko, Piotr Kurp, Tetyana I. Zhylenko, Ievgen Konoplianchenko

**Affiliations:** 1Laser Research Centre, Faculty of Mechatronics and Mechanical Engineering, Kielce University of Technology, Al. Tysiąclecia P.P. 7, 25-314 Kielce, Poland; b.antoszewski@tu.kielce.pl; 2The Department of Applied Material Science and Technology of Constructional Materials, Sumy State University, R. Korsakov Str., 2, 40007 Sumy, Ukraine; gaponova@pmtkm.sumdu.edu.ua (O.P.G.); t.zhylenko@phe.sumdu.edu.ua (T.I.Z.); 3Technical Services Department, Sumy National Agrarian University, H. Kondratiieva Str., 160, 40021 Sumy, Ukraine; tarelnyk@ukr.net (V.B.T.); yevhen.konoplianchenko@snau.edu.ua (I.K.); 4Department of Physical Chemistry of Inorganic Materials, Frantsevich Institute for Problems of Materials Science, Krzhizhanovsky Str. 3, 03142 Kyiv, Ukraine; zvyagina47@gmail.com

**Keywords:** electrospark alloying, coatings, microhardness, continuity, roughness, structure, X-ray diffraction analysis, X-ray spectral analysis

## Abstract

In this paper, the possibility of applying the electrospark alloying (ESA) method to obtain boron-containing coatings characterised by increased hardness and wear resistance is considered. A new method for producing such coatings is proposed. The method consists in applying grease containing aluminium powder and amorphous boron to the surface to be treated and subsequently processing the obtained surface using the ESA method by a graphite electrode. The microstructural analysis of the Al-C-B coatings on steel C40 showed that the surface layer consists of several zones, the number and parameters of which are determined by the energy conditions of the ESA process. Durametric studies showed that with an increase in the discharge energy influence, the microhardness values of both the upper strengthened layer and the diffusion zone increased to W_p_ = 0.13 J, Hµ = 6487 MPa, and W_p_ = 4.9 J, Hµ = 12350 MPa, respectively. The results of X-ray diffraction analysis indicate that at the discharge energies of 0.13 and 0.55 J, the phase composition of the coating is represented by solid solutions of body-centred cubic lattice (BCC) and face-centred cubic lattice (FCC). The coatings obtained at W_p_ = 4.9 J were characterised by the presence of intermetallics Fe_4_Al_13_ and borocementite Fe_3_ (CB) in addition to the solid solutions. The X-ray spectral analysis of the obtained coatings indicated that during the electrospark alloying process, the surface layers were saturated with aluminium, boron, and carbon. With increasing discharge energy, the diffusion zone increases; during the ESA process with the use of the discharge energy of 0.13 J for steel C40, the diffusion zone is 10–15 μm. When replacing a substrate made of steel C40 with the same one material but of steel C22, an increase in the thickness of the surface layer accompanied by a slight decrease in microhardness is observed as a result of processing with the use of the ESA method. There were simulated phase portraits of the Al-C-B coatings. It is shown that near the stationary points in the phase portraits, one can see either a slowing down of the evolution or a spiral twisting of the diffusion-process particle.

## 1. Introduction

Most critical parts of compressors, pumps, gas transfer apparatuses, and other dynamic equipment (DE) operate at high speeds, pressures, and temperatures, and under abrasive and corrosive conditions, or exposure to other types of working environments. Usually, the problems associated with improving the reliability and durability of those parts are solved through the use of expensive and high-hardness materials. In [1], it was shown that in high-speed pumps and high-pressure compressors, non-contact type mechanical seals are widely used, whose sealing rings are made entirely of wear-resistant materials such as tungsten carbide, silicon carbide, and various types of graphite. The cost of the rings made of these materials reaches hundreds and thousands of US dollars, with the attendant high cost of the sealing units as a whole [2]. Being accompanied by an increase in operating parameters of the DE, the development of technology entails the need for the appearance of new and cheaper but no less reliable composite materials that combine the protective properties of the coatings with the mechanical strength of the substrate. Therefore, investigations aimed at creating such materials as a “base-coating” type are relevant and timely.

Surface engineering has a large number of technologies allowing the formation of the surface layers with special properties. In industry, among the most widely used methods is chemical and thermal treatment (CTT); of particular note is the use of traditional technologies such as carburising, nitriding, and nitrocarburising, which are associated with their sufficient coverage of studies and relative simplicity. It should be noted that such CTT methods as chromising, boriding, aluminising, and their combinations (carboboriding, borochromising, boroaluminising, etc.), despite the fact that they often significantly surpass traditional methods, are less in demand by technologists and heat-treaters. Their application is often limited by insufficient knowledge of the process technology, a lack of prediction of the final result (structure, hardness, warpage, residual stresses, etc.), and higher operational equipment requirements for the necessary equipment, sometimes associated with increased cost, etc. [3,4]. Boriding is one of the most effective processes for increasing the wear resistance of steel parts. As shown in [5,6,7,8], it is carried out by the CTT method including a diffusion saturation procedure in powders, gas media, molten salts, and electrolyte melts. Each of them has its own advantages and disadvantages. According to the authors of [9], among the existing boriding processes, taking into account their technological advantages and disadvantages, namely productivity (saturation rate) and cost (economic efficiency), the most preferred process is electrolysis borating, which in comparison with solid borating in powders requires neither special preparatory work nor preparation of powder mixtures. The introduction of the boriding process was carried out by firms: Degussa, Leybold Durferrit, Sandvik AB, Stahlwerke Röchling-Burbach, HEF, and others [9]. Using the boriding procedure, it is possible to achieve an increase in wear resistance by 3–50 and 1.5–15 times in comparison with heat treatment and traditional CTT methods, respectively [10]. In addition, diffusion boriding up to 5 times increases the heat resistance of steels [3]. Boride coatings have the highest wear resistance together with good heat and corrosion resistance in comparison with widely used nitriding and carburising practices. Despite the indisputable advantages of the boriding process carried out by the CTT method that makes it possible to increase surface layer hardness, wear resistance, corrosion resistance, etc., the possibility of a surface layer embrittlement cannot be excluded, which is the main disadvantage of the boride layers. In [11], it is stated that the increased fragility and tendency to form cracks and chips are explained by the anisotropy of the thermal expansion of the boride phases (FeB and Fe_2_B). The absolute values of the thermal expansion coefficients of the diffusion layer phases as well as the basics and the natures of their changes depending on temperature values affect the size and distribution of the temporal and residual stresses by the layer depth. The solution to the problem of reducing the embrittlement of the boride layer can be carried out in various ways, including alloying it, performing a diffusion layer with a maximum Fe_2_B phase content, and providing the formation of a single-phase layer. As a result of brittleness studies of the layers performed on the base of refractory metal borides (of titanium, vanadium, chromium, and tungsten) obtained by the synthesis of the corresponding oxides in a vacuum using electron-beam heating (electron-beam boriding) [12,13], it has been found that a process for forming refractory metal borides (of titanium, vanadium, chromium, and tungsten) in a layer applied can reduce its brittleness and increase the ductility thereof. A large number of modern studies have been dedicated to finding solutions for obtaining high-quality boride coatings by improving technology [14], as well as the composition of the boride coating [15,16].

Alternative technologies for producing the boride coatings, which have none of the inherent disadvantages of the CTT (process time, low productivity, the need to heat the entire workpiece, deformation and warping, the need for expensive equipment, etc.), are the boriding methods using concentrated energy fluxes (CEFs) [17,18,19,20]. In [21], a method is proposed for electroerosive boriding including applying a boron-containing surface coating by an electrode made in the form of a rod blown with a cooler. Ferroboron is used as a boron-containing electrode, and compressed air or a neutral gas is used as a cooler.

The application of local surface treatment procedures based on the method of electrospark alloying (ESA) would make it possible to obtain the layers with the desired properties. The ESA method advantages are locality, low energy consumption, the absence of volumetric heating of the material, high adhesion of the coating with the base metal, the ease of automation of the process, environmental safety, and economy [22,23]. The ESA method allows for providing the implementation of the CTT processes such as carburising, aluminising, and chromising. Additionally, with the ESA method, not only the elements of the conductive material of the anode (consisting of, for example, metal or graphite) but also dielectrics (for example, sulphur, boron, etc.) can be introduced into the surface of the part by putting them into the liquid micro-bath formed when local electric spark discharges occur. We have shown that the alloying procedure performed by the elements during the ESA process can be accomplished by creating a special technological environment (STE) without the use of expensive electrode materials and with a reduction in the time for obtaining a coating, thereby increasing productivity and cutting production costs. This can likewise be done for processes such as sulphidising [24], sulfocarburising, sulfoalitising [25], and complex coatings [26]. While considering the ESA process, it becomes clear that an important role is played by the method for thermodynamic modelling of coating synthesis, which makes it possible to evaluate the interaction products profile in the multiphase multicomponent systems [27].

The physical nature of the electrospark process performed with the influence of energy fluxes is a complex of complicated mechanical, physical, and chemical processes occurring on the contact surface in the volume of the work piece surface layer. These processes are associated with such phenomena as deformation, friction, physicochemical transformations, adhesive events, heat transfer, energy dissipation, cavitation, wave processes, material wear and tear, etc. They have a significant impact on the work piece, and as a whole, lead to a qualitatively new behaviour of the technological system (TS). The ESA processes are the complex TSs wherein there are possible self-organising processes. In the scientific literature, there is a relatively small number of works devoted to the study of CEF processing in terms of synergetics. Among the few papers, it is worth noting [28,29,30], wherein the mechanisms of evolution and self-organisation in the interconnected and interdependent physical processes accompanying the surface treatment procedures are examined. An analysis of those works makes it possible to draw a conclusion, on the one hand, about the prospects for studying CEF processing in terms of synergetics and, on the other hand, about insufficient knowledge of the mechanisms of self-organisation and evolution in the TS for CEF processing, especially relative to the ESA method. Despite the positive results achieved by the synergetic approach to such processes, to date there is no single theory that comprehensively describes the processes of self-organisation in the TS of the spark alloying method.

When analysing the ESA processes, a researcher has to deal with a complex physical system, which is characterised by dynamic and stochastic components that lead to self-organisation. It is possible to predict its behaviour through the use of the modified Lorentz system, which is intended for work with such components [31]. It is a common fact [32] that the Lorentz system consists of a field, an order parameter, and a control parameter. Using the modified Lorenz system, it is possible to optimise and predict the quality characteristics and the parameters of the coatings. The conditions under which the processes that determine the ESA procedures occur are non-equilibrium. The author of [33], as well as those of [34], have shown that in these cases, there may be present such phenomena as alloying electrode erosion, development of anode mass flow, the coating formation, and the formation of an altered surface layer. Therefore, it is relevant to apply the synergistic laws to the ESA procedures based on the study of phase transitions of the process dynamic components using the phase plane method.

The aim of this work is to improve the quality and environmental safety of the boron technology through the development of a new method for producing Al-C-B system coatings with the use of the ESA method; through studying the process of a surface layer structure and phase formation, depending on the energy parameters of the equipment used; and through studying the kinetics of the process for the surface layer formation on a steel substrate using the phase plane method.

## 2. Methodology

Samples of steel C22 and steel C40 (METINVEST HOLDING LLC, Mariupol, Ukraine) with dimensions of 15 × 15 × 8 mm were used. To the samples there was applied a grease consisting of mineral paste as a binder; PAD-0 (Dnepr-titan, Dnipro, Ukraine) grade aluminium powder, 56% by weight; and boron amorphous powder (Ukrsplav LTD, Dnipro, Ukraine) of about 5% by weight. Without waiting for the drying of the mineral paste, the surfaces of the samples were processed using the ESA method with the EG-4 (TK PROMEL, Kyiv, Ukraine) grade graphite electrode at the Elitron-52A model unit, applying a discharge energy of 0.13, 0.55, and 4.9 J.

The surface roughness after processing was determined with the use of the 201 model of profilograph profilometer manufactured at the Caliber plant by reading and processing the profilograms. The metallographic analysis of the coatings was carried out using the Neophot-21 optical microscope (Carl Zeiss AG, Oberkochen, Germany).

Microhardness testing was performed using the PMT-3 instrument (LOMO, St. Petersburg, Russia) according to standard methods (GOST 9450-76). The test load was 50 g. The quantitative assessment of the experimental data of microhardness measurement was carried out by the methods of mathematical statistics. The normal distribution of microhardness of a series of samples hardened in the same mode was checked. The reproducibility of the experimental results was checked using Cochran’s criterion. Thus, a series of samples was investigated, and hardness measurements were carried out in several fields of a microsection. The microhardness distribution is the result of the processed data. The photos of microstructures (Section 3) show that all zones of the obtained coatings were measurable. The applied load did not lead to deformation or strengthening of the adjacent indentation area by cold work. 

To study the distribution of the elements by the depth of the layer, a local X-ray microanalysis was performed. Used in doing so was the scanning electron microscope model SEO-SEM Inspect S50-B equipped with the energy dispersive spectrometer model AZtecOne with the X-MaxN20 detector (Oxford Instruments plc, Abingdon-on-Thames, UK). The X-ray investigations were carried out in CoKα radiation using the DRON-UM1 diffractometer (voltage and current of 35 kV and 30 mA, respectively (Burevestnik, St. Petersburg, Russia). The diffraction patterns were read by the step-by-step scanning method. The scanning step was 0.050, and the exposure time at a point was 3 s.

The diffraction patterns were processed using the program for the full-profile analysis of X-ray spectra from the mixture of the PowderCell 2.4 polycrystalline components.

## 3. Results and Discussions

The research results indicate that the microstructure of the Al-C-B coatings consists of several zones, the number and parameters of which are determined by the energy conditions of the ESA process (Figure 1). At relatively low discharge energies (0.13 and 0.55 J), the layers consist of 3 zones. Those are the upper “white” strengthened layer, the diffusion zone, and the base metal, that is, steel C40 with a ferrite-pearlite structure (Figure 1a,b). Moreover, the size of the “white” layer for the above modes was 15–20 μm (Table 1). The increase of the discharge energy up to 4.9 J led to a change in the number of zones and their structures (Figure 1c): the upper layer with a dendritic structure (up to 60 μm); the sublayer (up to 20 μm); and the diffusion zone characterised by improved structural components. In this regard, it had an increased etchability in a reagent as well as the base material.

Durametric studies showed that with an increase in the discharge energy, the microhardness of both the upper strengthened layer and the diffusion zone increased (Figure 2), at W_p_ = 0.13 J, Hµ = 6487 MPa, and at W_p_ = 4.9 J, Hµ = 12350 MPa (Table 1), respectively.

The results of X-ray diffraction analysis presented in Figure 3 indicate that at relatively low discharge energies (0.13 and 0.55 J), the phase composition of the coatings is represented by solid solutions of BCC (body-centred cubic lattice) and FCC (face-centred cubic lattice) with the parameter of a = 28.651 nm and 36.189 nm, respectively (Table 2). Therefore, it can be assumed that while steel C40 is saturated with Al, C, and B by the ESA method, as a result of mixing the base material, the mineral paste containing aluminium powder, the electrode material, and graphite, the BCC solid solution is alloyed and its parameter increases. In addition, accelerated cooling after the ESA process leads to the formation of thermal stresses, resulting in additional increase in the microhardness of the coating.

In addition to the solid solutions of BCC and FCC with the crystal lattice parameters altered upward (Table 2), the coatings obtained at W_p_ = 4.9 J are characterised by the presence of Fe_4_Al_13_ intermetallic compounds and Fe_3_(CB) borocementite. The formation of those phases contributes to significantly strengthening and increasing the microhardness of the surface layer up to 12350 MPa (Table 1).

The X-ray microanalysis of the obtained coatings indicates that during the electrospark alloying process, the surface layers are saturated with aluminium, boron, and carbon. With increasing discharge energy, the diffusion zone of these elements increases. So, at the ESA process with a minimum investigated discharge energy of 0.13 J for steel C40, the diffusion zone is about 12 μm.

When replacing a substrate made of steel C40 with the same one but of steel C22, as a result of processing with the use of the ESA method, an increase in the thickness of the surface layer accompanied by a slight decrease in microhardness is observed (Figure 2 and Figure 4). Obviously, the influence of carbon content in material (steel C22 contains about 0.2% carbon, steel C40 about 0.4%) is due to the formation of a multiphase hardening structure. Figure 5 shows microstructure of the Al-C-B and Figure 6 shows the results of electron microscopic studies of Al-C-B coatings on steel C22. At W_p_ = 0.13 J, thin and discontinuous layers are formed. With increasing discharge energy, the thickness of the coatings and their continuity increases.

It should be noted that with increasing discharge energy, the diffusion zone of Al, C, and B also increases in the coating. So, at W_p_ = 0.13 J, this zone is 5–7 μm, while at W_p_ = 4.9 J it is 23–25 μm (Figure 7). Carbon and aluminium diffuse to a greater extent far below the surface.

## 4. Kinetics of the Formation of Al-C-B Coatings by the ESA Method

As previously noted [32], the ESA process is ensured by a self-consistent change in the concentration of atoms n, stress σ, and substrate temperature T measured by ambient temperature. The process of increasing the thickness of the forming coating is conditioned by the fact that temperature T increases due to the formation of an excess of the atomised substance atoms. This enhances the evaporation of the precipitated substance atoms due to an increase in the absolute value of the internal stress σ < 0, which compensates for the initial concentration. The evolution equations of these quantities contain a dissipative contribution and terms representing the positive and negative feedbacks shown in the modified dimensionless Lorentz system [33]:τ·n’ = −n·σ + ε_n_·ξ,(1)
τ_T_·T’ = −T + n·σ + ε_T_·η,(2)
τ_σ_·σ’ = (σ_0_ − σ) − n·T,(3)
where in:ε_n_, ε_T_—concentration fluctuation amplitude and temperature fluctuation amplitude;ξ, η—independent white noise;τ_T_—temperature relaxation time;τ_σ_—stress relaxation time;τ—concentration relaxation time;
where the relaxation time is defined as
ξ = ξ(t);  〈ξ(t)〉 = 0;  〈ξ(t)ξ(t’)〉 = δ(t − t’)(4)

Based on the phase plane method, we will discuss the evolution of the system in more detail assuming, in turn, that one of the degrees of freedom has the highest relaxation rate [34].

We will first consider the behaviour of the system at rapid stress relaxation, τ_σ_ << τ_T_ = τ:τ·n’ = −n + σ_0_ − n·T + ε_n_·ξ,(5)
 τ_T_·T’ = −T + n·(σ_0_ − n·T) + ε_T_·η (6)

In the phase portraits (Figure 8, Figure 9 and Figure 10), there can be observed only the attracting part (unit), which corresponds to the steady state characterising the diffusion process.

At low values of internal stress and concentration, a single state is realised corresponding to the stationary mode of condensation. Comparison of phase-plane portraits corresponding to different ratios of relaxation times shows that the fastest evolution of the system occurs along the axis corresponding to the shortest of these times. In this case, near the stationary points, either a zone of slow evolution (the channel of a large river), or a spiral zone, on which a non-monotonic mode of condensation is realised, is observed. A study of the conditions for the realisation of this regime shows that it is favoured by an increase in the temperature relaxation time. Thus, phase-plane portraits with the specified range of temperature, stress, and concentration changes make it possible to form a specialist’s idea of the duration of the formation of the diffusion layer and the parameters that affect the appearance of new layers.

Fast relaxation of temperature, τ_T_ << τ_σ_ = τ:τ·n’ = −n + σ + ε_n_·ξ,(7)
τ_σ_·σ’= (σ_0_ − σ) − n·(n·σ + ε_T_·η) (8)

Fast relaxation of concentration, τ << τ_σ_ = τ_T_:τ_T_·T’ = −T + σ·(σ + ε_n_·ξ) + ε_T_·η,(9)
 τ_σ_·σ’ = (σ_0_−σ) − T·(σ + ε_n_·η) (10)

## 5. Summary and Conclusions

1. The possibility of applying the ESA method to obtain boron-containing coatings characterised by increased hardness and wear resistance was considered.

2. The microstructural analysis of the Al-C-B coatings showed that the surface layer consists of several zones, the number and parameters of which are determined by the energy conditions of the ESA process. Durametric studies showed that with an increase in the discharge energy influence, the microhardness values of both the upper strengthened layer and the diffusion zone increase.

3. The results of X-ray diffraction analysis indicate that at the discharge energies of 0.13 and 0.55 J, the phase composition of the coating is represented by solid solutions of BCC (body-centred cubic lattice) and FCC (face-centred cubic lattice) with the parameter of *a* = 28.651 nm and 36.189 nm, respectively. In addition to the solid solutions of BCC and FCC, the coatings obtained at W_p_ = 4.9 J are characterised by the presence of intermetallics Fe_4_Al_13_ and borocementite Fe_3_(CB), with the altered upward parameters of the crystal lattice. The formation of these phases contributes to a significant strengthening and increasing of the microhardness of the surface layer up to 12,350 MPa.

4. The X-ray microanalysis of the obtained coatings indicates that during the electrospark alloying process, the surface layers are saturated with aluminium, boron, and carbon. With increasing discharge energy, the diffusion zone increases.

5. There were simulated phase portraits in the MathCAD environment. The kinetics of the layer formation processes were investigated at different ratios of the relaxation time of the sprayed material, the internal stress, and the temperature of the substrate. It was shown that near the stationary points in the phase portraits, one can see either a slowing down of the evolution or a spiral twisting of the diffusion process particle. Having analysed an image, a specialist can possess information about the limits for changing the values of the temperature or atom concentrations on the surface and thus be enabled to control the process of obtaining boron-containing coatings of the Al-C-B system by the ESA method.

## Figures and Tables

**Figure 1 materials-14-00739-f001:**
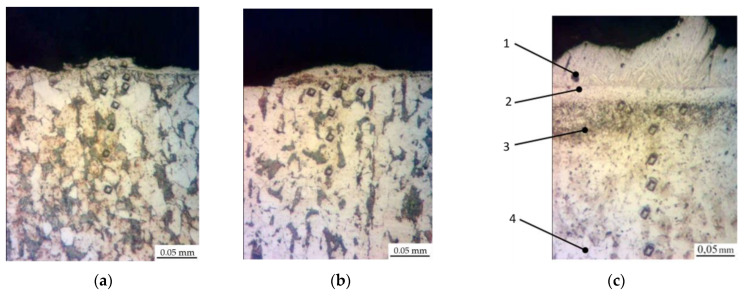
Microstructures of the Al-C-B coatings on steel C40: (**a**) W_p_ = 0.13 J; (**b**) W_p_ = 0.55 J; (**c**) W_p_ = 4.9 J. The following zones can be identified: **1**.—“white” layer, **2**.—sublayer, **3**.—diffusion zone, **4**.—base material.

**Figure 2 materials-14-00739-f002:**
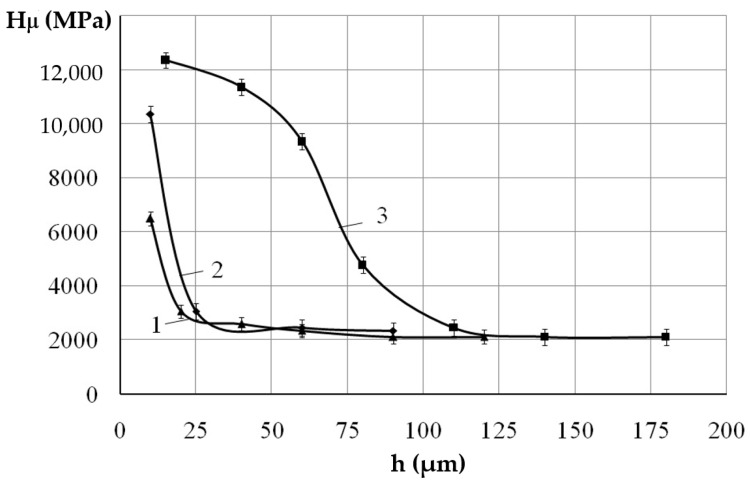
Microhardness distribution of the Al-C-B coatings on steel C40: (**1**) W_p_ = 0.13 J, (**2**) W_p_ = 0.55 J, (**3**) W_p_ = 4.9 J.

**Figure 3 materials-14-00739-f003:**
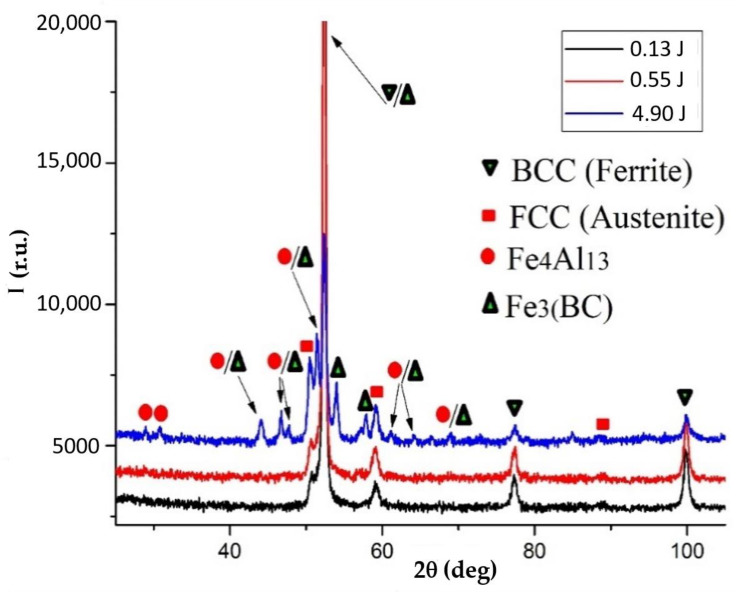
Diffraction patterns of Al-C-B coatings obtained by the electrospark alloying (ESA) method on steel C40.

**Figure 4 materials-14-00739-f004:**
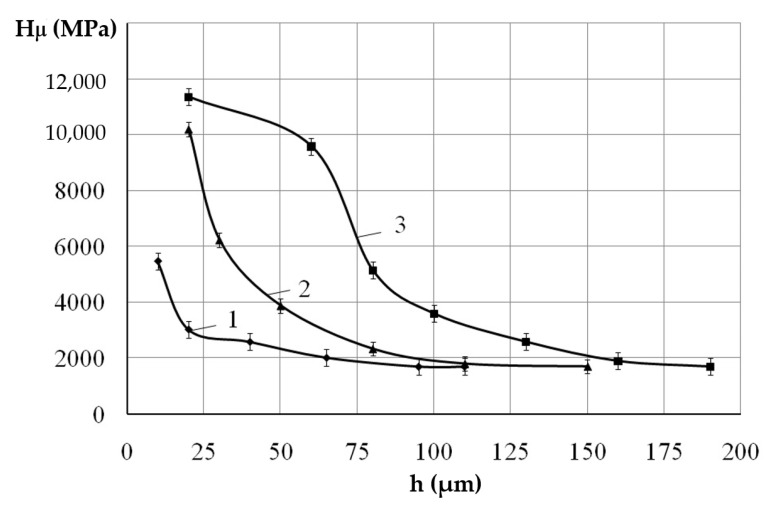
Microhardness distribution for Al-C-B coatings on steel C22: (**1**) W_p_ = 0.13 J; (**2**) W_p_ = 0.55 J; (**3**) W_p_ = 4.9 J.

**Figure 5 materials-14-00739-f005:**
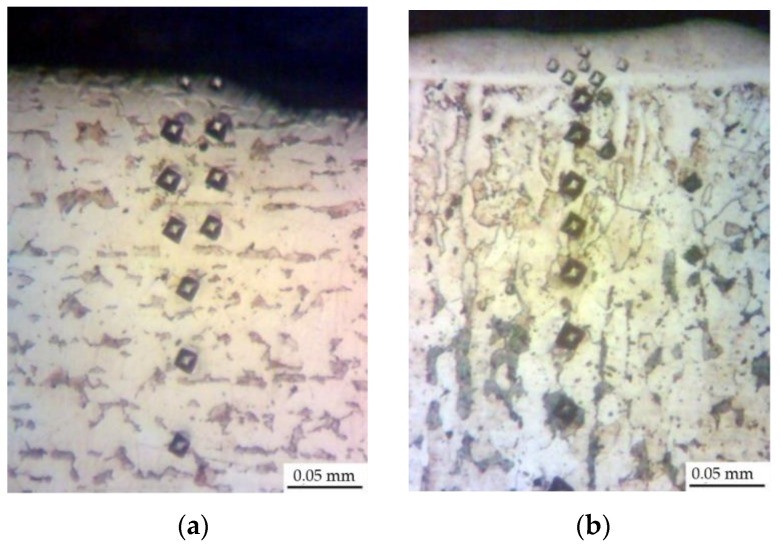
Microstructures for Al-C-B coatings on steel C22: (**a**) W_p_ = 0.55 J; (**b**) W_p_ = 4.9 J.

**Figure 6 materials-14-00739-f006:**
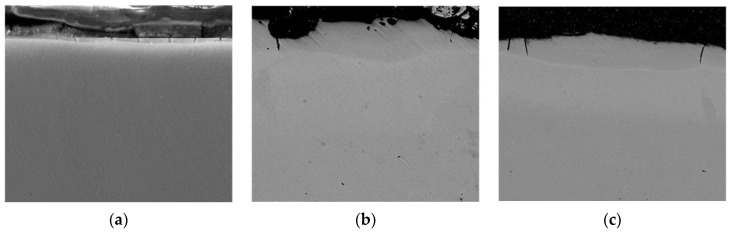
Structures of the surface Al-C-B coatings on steel C22 obtained by the ESA method: (**a**) W_p_ = 0.13 J (mag. 2400×); (**b**) W_p_ = 0.55 J (mag. 2400×); (**c**) W_p_ = 4.9 J (mag. 1200×).

**Figure 7 materials-14-00739-f007:**
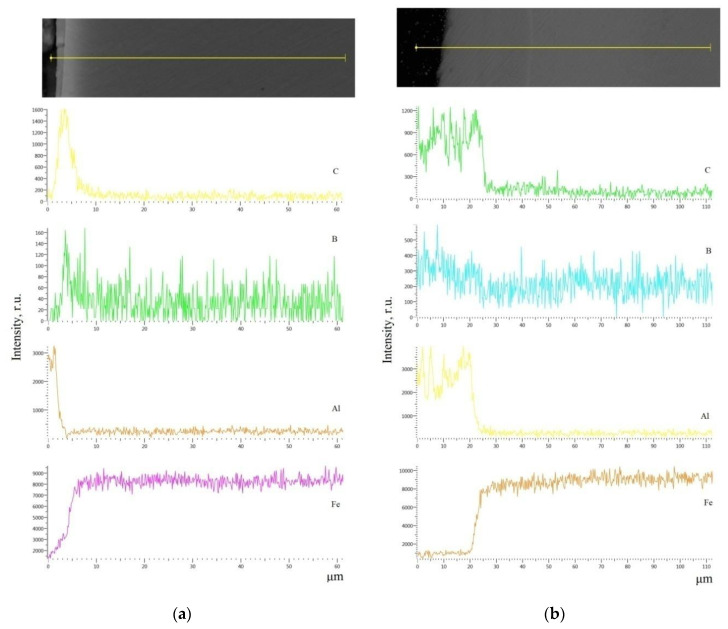
Distribution of the elements (carbon, boron, aluminium, and iron) in the Al-C-B coatings obtained by the ESA method with graphite electrode used on steel C22: (**a**) W_p_ = 0.13 J; (**b**) W_p_ = 4.9 J.

**Figure 8 materials-14-00739-f008:**
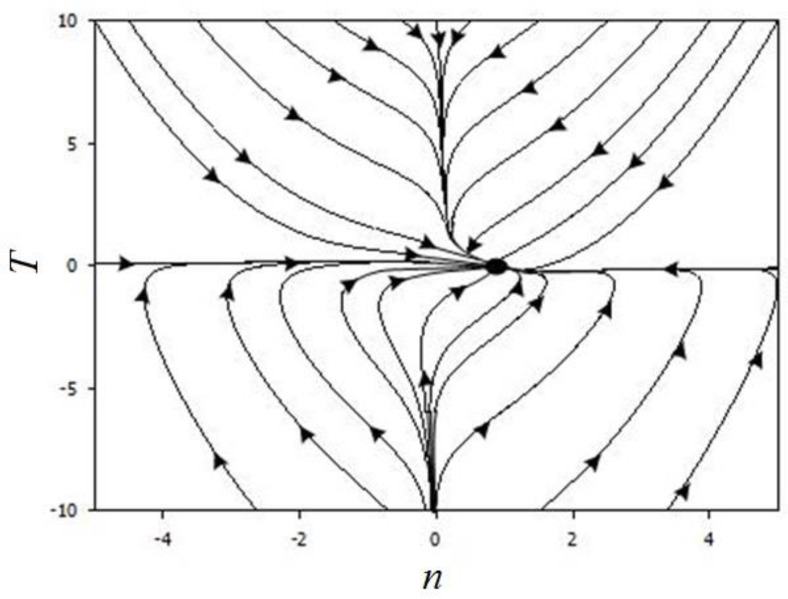
Phase portrait at τ_σ_ << τ_T_ = τ, σ_0_ = 0.5, τ_T_/τ = 1, ε_n_ = 0.5, ξ = 0.1, ε_T_ = 0.01, and η = 0.05.

**Figure 9 materials-14-00739-f009:**
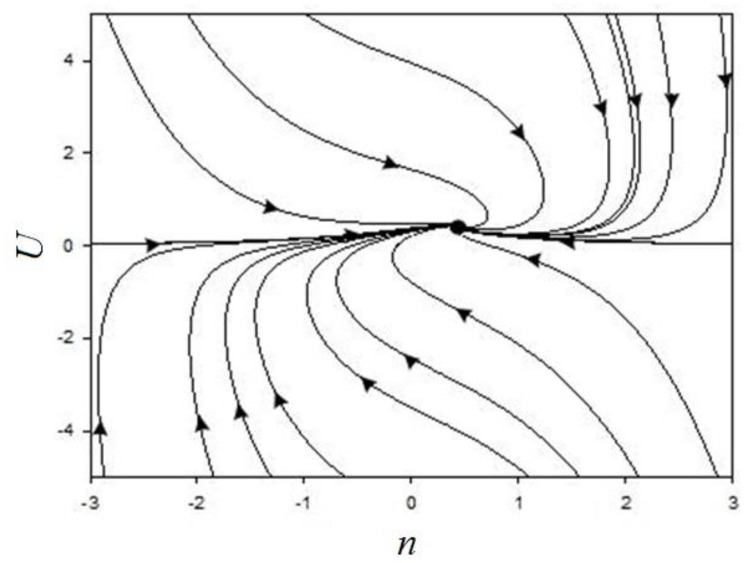
Phase portrait at τ_T_ << τ_σ_ = τ, σ_0_ = 0.5, τ_σ_/τ = 1, ε_n_ = 0.5, ξ = 0.1, ε_T_ = 0.01, and η = 0.05.

**Figure 10 materials-14-00739-f010:**
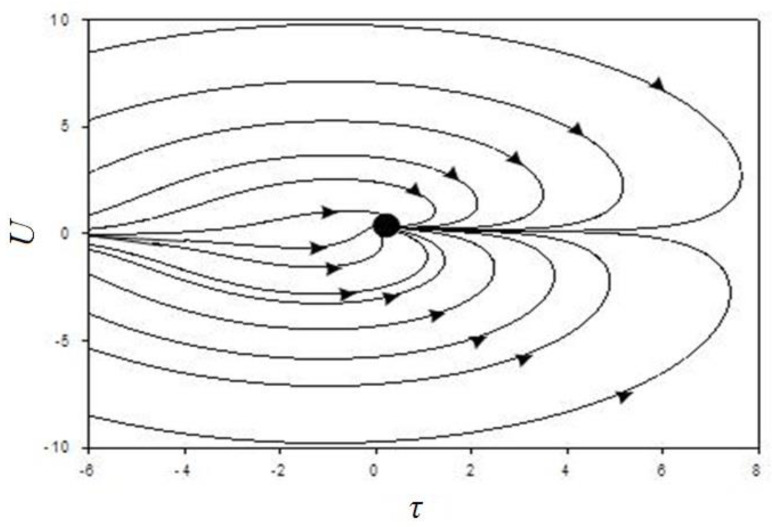
Phase portrait at τ << τ_σ_ = τ_T_, σ_0_ = 0.5, τ_σ_/τ_T_ = 1, ε_n_ = 0.5, ξ = 0.1, ε_T_ = 0.01, and η = 0.05.

**Table 1 materials-14-00739-t001:** Qualitative parameters of Al-C-B coatings obtained by the ESA method.

Discharge Energy, J	Roughness, μm			Strengthened Layer		
	Ra	Rz	Rmax	Hµ, MPa	h, μm	S, %
**Steel C40**						
0.13	1.2	2.9	7.4	6487	15	55
0.55	2.9	4.5	17.3	10351	20	75
4.9	9.3	19.5	48.2	12350	60	95
**Steel C22**						
0.13	1.1	2.7	7.2	5474	20	60
0.55	2.9	4.1	16.3	10196	30	80
4.9	8.9	18.7	46.1	11345	75	95

S,%—layer structural integrity.

**Table 2 materials-14-00739-t002:** Parameters of the crystal lattices of the phases and quantitative phase analysis of Al-C-B coatings on steel C40.

Discharge Energy, J	Phase	Parameters of Crystal Lattices, *a*, nm	Phase Content, % (mass.)
0.13	FCC solid solution	36.189	8
	BCC solid solution	28.651	92
0.55	BCC solid solution	36.189	8
	BCC solid solution	28.651	92
4.9	Fe_4_Al_13_	*a* = 153.920 *b* = 81.779 *c* = 124.914 *β* = 107.3709	10
	Fe_3_(CB)	*a* = 50.818 *b* = 67.791 *c* = 45.170	43
	FCC solid solution	*a* = 36.209	14
	BCC solid solution	*a* = 28.699	33

## Data Availability

Data sharing not applicable.

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
