# Peer review of "Assessment of Technological Capabilities for Forming Al-C-B System Coatings on Steel Surfaces by Electrospark Alloying Method"

_materials, 2021, doi:10.3390/ma14040739_

Round 1
Reviewer 1 Report
In this research, electrospark alloying method was proposed to obtain boron-containing coatings. It lacks innovation, and data analysis is too simple. Moreover, the coating does not show obvious superior performance. I suggest that this manuscript should be rejected.
Here are some more details:
- The paper suffers from major English language flaws: the manuscript would need a, preferably external, thorough linguistic check to ensure its grammatical, stylistic and typographical appropriateness.
- Which standard did the author follow in durametric studies? The experimental part of the article lacks detailed description.
- In Figure 1, 3 sections layers is hard to find in the image. Besides, it is hard to identify the ferrite-pearlite structure. The author's description goes beyond what the picture can actually convey
- What is the load on the hardness test? From Figure 2, the distance between two adjacent points is too small, which is easy to cause work hardening and lead to a high hardness value.
- There is an error in the captain of Figure 4.
Author Response
Thank You for Your comments, we attached our replies

Reviewer 2 Report
Dear Authors,
The manuscript focuses on production of novel coating method using Al-C-B system implementing electrospark alloying. The work should be deeply revised according the following remarks:
- English must be improved over the text. Some examples for the sentences which are not used in technical English:
Line 161: “Without waiting for drying the mineral jelly”.
Line 171: “In doing so”.
Line 261, 265 and 268: “Phase portrait” what does it mean?
- Line 41: “Environment Exposure” please write with small letters.
- You have mentioned wear resistant materials such as tungsten carbide, silicon carbide, various types of graphite. Please add some references related to their applications.
- Line 54: “layers with special and special properties”, please rewrite.
- Please provide the names of the manufacturers of the initial materials.
- Please add the electrical parameters of the XRD analysis (current and voltage).
- Please provide microhardness experimental description in Materials and Methods.
- Please show on Figure 1 the sections you have mentioned in the text. Unfortunately, these sections are hard resolvable.
- All patterns of XRD should be in one Figure and not separated to three, please rearrange. Moreover, the crystal lattices of BCC and FCC were not related to any phase, please specify what phase is this.
- Figure 6 presents SEM images of the obtained coatings. However, deeper analysis is needed to improve the quality of the research. I recommend going for the higher resolutions and to detect any structural changes.
- Conclusions are too long, please rewrite with the main achievements.
- 15 from 32 references are in Russian or Ukrainian language. Please find alternative reference and change them. The should be not more than 3-4.
Author Response
Thank You for Your comments, there are our replies:
- English must be improved over the text. Some examples for the sentences which are not used in technical English:
Line 161: “Without waiting for drying the mineral jelly”.
Line 171: “In doing so”.
Line 261, 265 and 268: “Phase portrait” what does it mean?
Our reply:
Full test was check by native speaker.
We change "mineral jelly" formulation to "mineral paste" formulation.
Partly agree with remark 1. Notion “Phase portrait”used in technical English. For example, [Martín-Antonio Rodríguez-Licea, Francisco-J. Perez-Pinal, José-Cruz Nuñez-Pérez On the n-Dimensional Phase Portraits, Appl. Sci. 2019, 9, 872. doi:10.3390/app9050872].
- Line 41: “Environment Exposure” please write with small letters.
Our reply:
Agree with Remark 2.
- You have mentioned wear resistant materials such as tungsten carbide, silicon carbide, various types of graphite. Please add some references related to their applications.
Our reply:
Agree with Remark 3. We made the necessary changes.
- Line 54: “layers with special and special properties”, please rewrite.
Our reply:
Agree with Remark 4.
- Please provide the names of the manufacturers of the initial materials.
Our reply:
Agree with Remark 5 (addesd in lines 70-71).
- Please add the electrical parameters of the XRD analysis (current and voltage).
Our reply:
Agree with Remark 6.We made the necessary changes. (added in lines 167-170)
- Please provide microhardness experimental description in Materials and Methods.
Our reply:
Agree with Remark 7. (added in lines 160-162)
- Please show on Figure 1 the sections you have mentioned in the text. Unfortunately, these sections are hard resolvable.
Our reply:
Partly agree with remark 8. The text contains explanations of sections Figure 1 (Line 178-183)
- All patterns of XRD should be in one Figure and not separated to three, please rearrange. Moreover, the crystal lattices of BCC and FCC were not related to any phase, please specify what phase is this.
Our reply:
Agree with Remark 9.We made the necessary changes.
- Figure 6 presents SEM images of the obtained coatings. However, deeper analysis is needed to improve the quality of the research. I recommend going for the higher resolutions and to detect any structural changes.
Our reply:
Partly agree with remark 10. Deeper analysis of the obtained coatings is not part of the research task. In the next work, we will definitely present these results.
- Conclusions are too long, please rewrite with the main achievements.
Our reply:
Agree with Remark 11.We made the necessary changes.
- 15 from 32 references are in Russian or Ukrainian language. Please find alternative reference and change them. The should be not more than 3-4.
Our reply:
Agree with Remark 12. We made the necessary changes.
Reviewer 3 Report
This manuscript reports the Al-C-B coatings on steel surfaces by elecrospark alloying (ESA) method. The authors characterized the coatings by the microstructure analysis, durametric studies, and X-ray diffraction measurements. The microstructure and structure of coating change with increasing discharge energies. The durametric studies have confirmed the enhancement of microhardness by the coating. The authors have also introduced simulated phase portraits.
There are enormous demands for cheap and reliable coating materials with high strength. The boron coating, which can achieve a high strength coating, is highly attractive. However, the surface layer is generally brittle. The authors have proposed that the ESA method might keep the hardness of the boron-based coating. Actually, the authors have succeeded in the enhancement of microhardness in Al-C-B coatings. I think that the manuscript meets all criteria necessary for Materials. But, before the acceptance, I recommend the authors to address the comments listed below to improve the readability of paper.
- In the abstract, the authors use abbreviated words without mentioning the meaning. I recommend the authors to give the meanings of the abbreviated words.
- In the introduction, there are many line breaks. To improve the readability, I recommend the authors to rewrite the introduction.
- In section 3, the authors should discuss the comparison between the simulations and the experimental results.
- In line 269, I think Figures 6-8 are Figures 8-10.
Author Response
Thank You for Your comments, there are our replies:
- In the abstract, the authors use abbreviated words without mentioning the meaning. I recommend the authors to give the meanings of the abbreviated words.
Our reply:
Agree with Remark. There are changes in the text
- In the introduction, there are many line breaks. To improve the readability, I recommend the authors to rewrite the introduction.
Our reply:
Agree with Remark. There are changes in the text
- In section 3, the authors should discuss the comparison between the simulations and the experimental results.
Our reply:
Agree with Remark. We made the necessary additions.
- In line 269, I think Figures 6-8 are Figures 8-10.
Our reply:
Agree with Remark. There are changes in the text
Reviewer 4 Report
1.Try to substantiate the claim that Boriding is one of the most effective processes for increasing the wear resistance of steel parts [65] mainly in connection with its disadvantages in the field of mechanical properties of these coatings.
2. How does the base material (its chemical composition, technological properties and mechanical properties) affect the changes in the mechanical properties of the formed layers. Try to justify the results in lines 300-302, not just state the result.
The scientific study is very interesting and beneficial both in terms of its content, results and methodology of experimental data processing. Apart from these two questions, I have no further comments on the authors.
Author Response
Thank You for Your comments, there are our replies:
1.Try to substantiate the claim that Boriding is one of the most effective processes for increasing the wear resistance of steel parts [65] mainly in connection with its disadvantages in the field of mechanical properties of these coatings.
Our reply:
Agree with Remark. We made the necessary additions. (lines 71-75)
- How does the base material (its chemical composition, technological properties and mechanical properties) affect the changes in the mechanical properties of the formed layers. Try to justify the results in lines 300-302, not just state the result.
Our reply:
We made the necessary additions in the text.
Round 2
Reviewer 1 Report
According to ASTM E384-17 “Standard Test Method for Microindentation Hardness of Materials”, the minimum recommended spacing between separate microindentation hardness tests is 2.5 times Vickers width, and the minimum distance between an indentation and the surface of the specimen is also 2.5 times Vickers width. However, as can be seen from the indentation in Fig. 1 and Fig. 2, the author did not follow the standard. In addition, the indentation area in Fig. 1c is significantly larger than that in Fig. 1a and b. Under the same load, the hardness values of steel C40 matrix calculated by the authors were the same, which can be seen in Fig. 2. So I doubt the accuracy of the data, and propose a revision of the hardness test.
Author Response
Thank you for your comments
There is our reply:
The quantitative assessment of the experimental data of microhardness measurement was carried out by the methods of mathematical statistics. The normal distribution of microhardnessof a series samples hardened in the same mode was checked. The reproducibility of the experimental results was checked using the Cochran's criterion. Thus, a series of samples was investigated, and hardness measurements were carried out in several (not one) fields of a microsection. The microhardness distribution shown in Fig. 2 is the result of the processed data. Regarding the remark related to the distance between indentation area(Fig. 1), we want to note that perhaps we presented a not very good photo. This photo we wanted to show that all zones of the obtained coatings were measurable.
Reviewer 2 Report
Dear Authors,
Thank you for revising the manuscript, however, I still have some issues:
- Remark 3: I’ll not check each reference, please summarize me what issue is discussed in which reference.
- Remark 5: You have to add the names of manufacturers of the materials you have used in the work in Methodology.
- Please add to the Methodology how many microhardness tests you have made.
- Remark 8: The text is OK, I have an issue with Figure 1. No sections are recognized, it is very nice that you have described them, but they are not seen on the image.
Author Response
There are attached reviews replies
